# A Collaborative Deprescribing Intervention in a Subacute Medical Outpatient Clinic: A Pilot Randomized Controlled Trial

**DOI:** 10.3390/metabo11040204

**Published:** 2021-03-30

**Authors:** Anissa Aharaz, Jens Henning Rasmussen, Helle Bach Ølgaard McNulty, Arne Cyron, Pia Keinicke Fabricius, Anne Kathrine Bengaard, Hayley Rose Constance Sejberg, Rikke Rie Løvig Simonsen, Charlotte Treldal, Morten Baltzer Houlind

**Affiliations:** 1The Capital Region Pharmacy, 2730 Herlev, Denmark; helle.bach.oelgaard.mcnulty@regionh.dk (H.B.Ø.M.); anne.kathrine.pedersen.bengaard.02@regionh.dk (A.K.B.); hayley.rose.constance.sejberg@regionh.dk (H.R.C.S.); rikke.rie.loevig.simonsen@regionh.dk (R.R.L.S.); ctreldal@gmail.com (C.T.); morten.baltzer.houlind@regionh.dk (M.B.H.); 2Multidisciplinary Outpatient Clinic (Fællesambulatoriet, subakutte patientforløb), Copenhagen University Hospital—Amager and Hvidovre, 2300 Copenhagen, Denmark; jens.henning.rasmussen@regionh.dk (J.H.R.); arne.cyron@regionh.dk (A.C.); 3Department of Clinical Research, Copenhagen University Hospital—Amager and Hvidovre, 2650 Copenhagen, Denmark; pia.keinicke.fabricius@regionh.dk; 4Department of Emergency Medicine, Copenhagen University Hospital—Bispebjerg and Frederiksberg, 2400 Copenhagen, Denmark; 5Department of Clinical Medicine, University of Copenhagen, 2200 Copenhagen, Denmark; 6Department of Drug Design and Pharmacology, University of Copenhagen, 2100 Copenhagen, Denmark

**Keywords:** subacute care, deprescribing, medication review, potentially inappropriate medication, STOPP, multimorbidity, hospital pharmacy service, ambulatory care facilities

## Abstract

Medication deprescribing is essential to prevent inappropriate medication use in multimorbid patients. However, experience of deprescribing in Danish Subacute Medical Outpatient Clinics (SMOCs) is limited. The objective of our pilot study was to evaluate the feasibility and sustainability of a collaborative deprescribing intervention by a pharmacist and a physician to multimorbid patients in a SMOC. A randomized controlled pilot study was conducted, with phone follow-up at 30 and 365+ days. A senior pharmacist performed a systematic deprescribing intervention using the Screening Tool of Older Persons’ potentially inappropriate Prescriptions (STOPP) criteria, the Danish deprescribing list, and patient interviews. A senior physician received the proposed recommendations and decided which should be implemented. The main outcome was the number of patients having ≥1 medication where deprescribing status was sustained 30 days after inclusion. Out of 76 eligible patients, 72 (95%) were included and 67 (93%) completed the study (57% male; mean age 73 years; mean number of 10 prescribed medications). Nineteen patients (56%) in the intervention group and four (12%) in the control group had ≥1 medication where deprescribing status was sustained 30 days after inclusion (*p* = 0.015). In total, 37 medications were deprescribed in the intervention group and five in the control group. At 365+ days after inclusion, 97% and 100% of the deprescribed medications were sustained in the intervention and control groups, respectively. The three most frequently deprescribed medication groups were analgesics, cardiovascular, and gastrointestinal medications. In conclusion, a collaborative deprescribing intervention for multimorbid patients was feasible and resulted in sustainable deprescribing of medication in a SMOC.

## 1. Introduction

Deprescribing is a term that covers dose reduction and discontinuation of inappropriate medications. It is an essential concept in the fight against inappropriate medication prescribing for multimorbid patients and a cornerstone in medication review [1,2,3]. The number of patients with multimorbidity (two or more conditions) in the Danish population is growing and accounts for approximately 7%. Moreover, the number among older patients above 80 years of age is about 29% [4]. Multimorbid patients are at particularly high risk for adverse drug reactions (ADRs) as up to 85% are exposed to polypharmacy (≥5 prescribed medications) and 30–85% of older multimorbid patients are treated with at least one potentially inappropriate medication (PIM) [4,5,6]. Furthermore, multimorbid patients are prescribed potentially lifelong medications, but often these medications are not reviewed regularly. These reviews are important considering changes in the health status, organ function, and/or lifestyle of patients [7,8,9,10,11]. 

Medication review performed by clinical pharmacists in hospitals have been proposed as a central part of the solution to optimize medication prescribing [12]. Many clinical guidelines, algorithms, and deprescribing lists act as supporting tools for clinicians in the medication prescribing process and are highly useful in the deprescribing process [9]. However, most published randomized controlled trials (RCT) on medication reviews as a single intervention (without a follow-up) have found little effect on hospital admission, readmission, and mortality [13,14,15]. One reason could be that completing medication reconciliation at sector crossings is complex, and medication changes are often incompletely transferred to the medication list on discharge [16,17]. This could be due to a lack of collaboration between pharmacists and physicians, resulting in insufficient implementation of suggested medication changes. Another possible reason for failure of the aforementioned trials to find an effect of medication reviews on hospital admission, readmission, and mortality may be due to patients’ health conditions being affected during acute admission and/or the physicians in acute settings wanting to avoid interfering with complex prescribing of medication from other medical specialties. A third potential issue is that patients or their relatives are not always involved in the medication review and are thus unaware that certain medications have been deprescribed [18], thereby increasing the risk of patients continuing to take deprescribed medication. 

Recent medication review intervention studies with patient involvement and follow-up in outpatient clinics have found promising optimization of patients’ medications through reduction of drug-related problems (DRP) [19]. Furthermore, studies have shown that collaboration between clinical pharmacists and physicians significantly increases prescribing appropriateness across the healthcare sector [20,21,22]. In recent years, medical outpatient clinics across medical specialties have been established in most Danish regions. These clinics are designed to carry out broad clinical assessment of patients, improve treatment, and reduce the number of acute admissions in patients with multimorbidity. Some of these medical outpatient clinics are organized in multidisciplinary teams, including physicians, nurses, pharmacists, and physiotherapists, while others are only based on physicians and nurses. These medical outpatient clinics act as essential links between the primary and secondary healthcare sectors, especially for multimorbid patients, and the setup offers easy patient follow-up [23]. The aim of this randomized controlled pilot study was to determine the feasibility of a collaborative deprescribing intervention for multimorbid patients by a clinical pharmacist and a physician with follow-up in a subacute medical outpatient clinic (SMOC).

## 2. Results

During the inclusion period, 76 patients were eligible for inclusion. Of these, 72 patients were included (95%), while four patients declined to participate. A further five patients were excluded during the study period due to hospitalization either directly from the SMOC or up to 14 days after study inclusion. A total of 67 patients (mean age 72.5 ± 12.3 years, 57% men) completed the study (Table 1). Both intervention (*n* = 34) and control (*n* = 33) groups were comparable at baseline with respect to gender, age, number of medications, and number of comorbidities (Table 1). Of the patients, 57% (*n* = 38) were referred to the SMOC due to either anemia, dyspnea, pain, hypertension, edema, or decline in physical function. The five most frequent comorbidities in the study group were cardiovascular disease (56 patients, 78%), pain conditions (47 patients, 65%), mental/neurological illness (20 patients, 28%), respiratory disease (18 patients, 25%) and diabetes (16 patients, 22%). Sixty-six patients (99%) had ≥1 diagnoses within the five most frequent comorbidities. In total, the five most frequent comorbidities accounted for 198 out of 295 (67%) of all comorbidities. There was no difference in the incidence of acute admissions between the control and intervention groups at 30, 90, and 180 days post inclusion (*p* ≥ 0.052). 

### 2.1. Feasibility of Deprescribing 

In total, 41 deprescribing recommendations were performed in the intervention group versus five in the control group. The pharmacist recommended deprescribing for 52 medications in the intervention group, 36 (69%) of which the physician implemented. The physician implemented an additional five changes for the control group. Table A1 and Table A2 give a detailed overview of each deprescribed medication, reason for deprescribing, and whether the deprescribing medications were sustained at 30 and 365+ days post inclusion. Of the 41 deprescribed medications, 24 (59%) were identified by the Screening Tool of Older Persons’ potentially inappropriate Prescriptions (STOPP) version 2 criteria, while 17 (41%) were identified by clinical guidelines and clinical observation. Eight (20%) of the deprescribed medications were due to side effects (Table A1).

Thirty days after inclusion, 29 medications were discontinued in the intervention group and five in the control group, while the number of dosage reductions made were eight and zero in the intervention and control groups, respectively. Nineteen patients (56%) in the intervention group had at least one deprescribed medication (56%), compared to four patients (12%) in the control group (*p* = 0.0002) (Table 2). 

### 2.2. Deprescribed Medications and Number of Medications at 30 Days Post Inclusion 

In total, five medications were deprescribed in the control group. Of these, four medications (80%) were regular, while one (20%) medication was pro re nata (P.R.N.). All five (100%) medications were discontinued. The medications in question were colchicine, folic acid, insulin, venlafaxine, and alogliptin. In the intervention group, 37 medications were deprescribed. Of these, 30 (81%) were regular medications, while 7 (19%) medications were P.R.N. In total, 29 of the 37 medications (78%) were discontinued, while 8 (22%) medications were reduced in dose. Table 3 shows the most frequent types of deprescribed medications in the intervention group. 

In the control group, there was no difference in the mean number of medications per patient at inclusion after leaving the SMOC and at 30 days post inclusion (10.5, 10.6, and 10.5) (*p* ≥ 0.26). For the intervention group, the mean number of medications was 9.3 (3.2) at inclusion, 8.6 (3.4) after leaving the SMOC, and 8.4 (3.2) at 30 days post inclusion. The mean number of medications in the intervention group was significantly lower after leaving the SMOC and at 30 days post inclusion (*p* ≤ 0.015). For the intervention group, no difference was found in the mean number of medications for patients after the first intervention and at 30 days post inclusion (*p* = 0.47). Finally, for the control group, the mean number of prescribed medications increased by 0.1 (1.6) per patient between inclusion and 30 days post inclusion. For the intervention group, the mean number of prescribed medications decreased by 0.8 (1.9) per patient from inclusion to 30 days post inclusion. The change in number of medications was significantly different between the two groups (*p* = 0.033). 

### 2.3. Deprescribed Medications Sustained at 365+ Days Post Inclusion 

In the intervention group, three patients died during follow-up, resulting in 31 patients with a total of 32 deprescribed medications who were accessible for follow-up at 365+ days post inclusion (Table A1). Of these, 31 of 32 (97%) deprescribed medications were sustained from 30 days post inclusion. One deprescribed zopiclone was reversed during this period. In the control group, one patient died during follow-up, resulting in 32 patients with a total of five deprescribed medications who were accessible for follow-up at 365+ days post inclusion (Table A2). Of these, 5 of 5 (100%) deprescribed medications were sustained from 30 days post inclusion. In total, 16 of 31 patients (51%) in the intervention group had at least one deprescribed medication sustained at 365+ days post inclusion, compared with 4 of 32 patients (13%) in the control group (*p* = 0.0010).

## 3. Discussion

In a pilot RCT study, we tested the feasibility of a collaborative deprescribing intervention for multimorbid patients within a SMOC. There were three primary findings that indicated feasibility. First, a significantly great proportion of patients in the intervention group (56%) compared to the control group (12%) had at least one deprescribed medications sustained at 30 days post inclusion. Second, in total, ≥97% of all deprescribed medications were sustained at 365+ days post inclusion. Third, 95% of patients eligible to participate in the study were included and randomized in the study, and 93% of these patients completed the study.

On average, 1.1 and 0.2 medications per patient were deprescribed in the intervention and control groups, respectively. We observed a significant reduction in the average number of medications in the intervention group between inclusion and 30 days post intervention. Moreover, the average number of medications was unchanged in the control group. The ranking of most frequently deprescribed medications started with analgesics, then cardiovascular medications, gastrointestinal medications, and sedatives.

### 3.1. Results in Context of Other Studies 

To our knowledge, this is the first study to demonstrate feasibility of a collaborative deprescribing intervention between a pharmacist and a physician in a SMOC. The physician implemented 69% of the recommended deprescribed medications by the clinical pharmacist. This is within the implementation range of 61–90% reported by other studies with collaboration between pharmacists and physicians [16,24,25]. Considering the subacute setup for patients with symptoms requiring treatment, our reported implementation rate of 69% indicates that collaboration between pharmacists and physicians is feasible. We assume that increasing this implementation rate would require multiple intervention times, more staff time for follow-up, and closer cooperation with general practitioners (GPs). 

Our deprescribing results are comparable with other studies dealing with deprescribing in different clinical settings. In a recent study from 2020 by Dharmarajan et al., geriatricians deprescribed medications for patients in both an outpatient clinic and two long-term care facilities. The patients in this study had a patient profile similar to ours. The results of the study showed on average 1.0 and 1.4 deprescribed medications in the outpatient clinic and long-term care facilities, respectively. Furthermore, they concluded that analgesics, nutritional supplements, lipid-lowering agents, antihistamines, and acid blockers had the highest deprescribing success [26]. Another recently published Dutch RCT investigated the effect of a pharmacist-led interdisciplinary medication review with follow-up in a cardiovascular outpatient clinic. The results showed a significant reduction in DRP of 0.8 and 0.3 for the intervention and control groups, respectively [19]. The authors highlighted that antihypertensives, antiarrhythmics, and analgesics were frequently involved in DRPs. Marvin and colleagues investigated the deprescribing process in older patients with acute admission for falls. They deprescribed on average 0.7 medications per patient, most commonly antihypertensives, opioids, sedatives, and nitrates [13]. Moreover, in a Danish longitudinal feasibility study by Houlind et al., 1.6 medications were on average deprescribed based on a pharmacist–geriatric medication review intervention in the emergency department. The most frequently deprescribed medications observed by Houlind et al. were proton pump inhibitors, analgesics, antihypertensives, and statins [16]. In contrast to our study, Marvin et al. and Houlind et al. found no difference in the total number of medications prior to intervention and during follow-up. For both studies, this was explained by new medications being prescribed during admission. Because the 365+ days post inclusion data was limited to patients with ≥1 deprescribed medications at 30 days post inclusion, we cannot determine whether the intervention had an impact on the total number of prescriptions after 365+ days. However, it is very likely given that 97% of deprescribed medications in the intervention group had their deprescribed status sustained at 365+ days post inclusion. 

Altogether, the results from our study are consistent with the literature and make it clear that it is often feasible and safe to deprescribe ≥1 medication for older multimorbid patients experiencing polypharmacy [16,24,26,27,28,29]. The results also emphasize that the same classes of medications are being deprescribed and/or causing DRPs across medical specialties and different clinical settings. This can be explained by the fact that the patients in the studies had many of the same comorbidities, including cardiovascular disease and pain conditions. Based on the aforementioned studies and the results of our study, it can be speculated that the type of setting, inpatient versus outpatient, has a minor impact on the number and type of medications deprescribed. This can be an important finding as the fight against inappropriate polypharmacy must be a consistent part of daily practice in many places in a coherent health system across sectors. 

Lundby et al. investigated the attitudes of healthcare professionals toward deprescribing in older patients and identified four important themes: (i) patient and relative involvement, (ii) teamwork, (iii) skills of healthcare professionals, and (iv) organizational factors [30]. Our study is based on a framework of all four themes, and its success is largely due to the well-established collaboration between clinical pharmacists and physicians. The GPs must be an active part of the focus on deprescribing, but we believe there is a lack of knowledge about deprescribing, collaboration, and time in daily practice to implement a nationally consistent effort. One study found that trained clinical pharmacists used approximately 25 minutes per intervention [25], which underlines why deprescribing is time-consuming for GPs in daily practice [17]. One solution might be for GPs and clinical pharmacists to cooperate on deprescribing, which has already been done and assessed in international studies on inappropriate prescribing [31]. Another realistic solution is using pharmacists in collaborative models in outpatient clinics in the fight against inappropriate polypharmacy. The results of our study indicate that this is a feasible solution.

### 3.2. Follow-Up, Shared Decision-Making, and Electronic Deprescribing Tools 

In our intervention group, the deprescribed status was sustained for 94% of the medications 30 days post inclusion. A Danish study found that only 64% of all prescribing changes in hospitalized geriatric patients were accepted by GPs [32]. The authors suggested that this low acceptance rate was related to miscommunication between the two healthcare sectors [33]. Studies have reported that involving patients in the describing process and keeping patients updated about their own medication status is just as crucial for implementation of deprescribing within different settings [34,35]. We believe that our high rate of sustained deprescribed medications is the result of utilizing motivational conversation together with following up with the patient seven days post inclusion by telephone. In this way, patients can become more involved in their own medical treatment. The patients in our study knew exactly why the changes were implemented, what they should be aware of in relation to the deprescribed medication, and what the goal of deprescribing was. The results from the Danish randomized clinical multicenter Odense Pharmacist Trial Investigating Medication Interventions at Sector Transfer (OPTIMIST) investigated the effect of an extended intervention. The extended intervention included a medication review with a motivational conversation and follow-up telephone call. The result was a significant reduction in readmissions among older hospitalized patients exposed to polypharmacy [25]. This result indicates the potential for involving and motivating patients in order to optimize medication prescribing. 

The recently developed version 2 of the STOPPFrail tool, which has been tested in a RCT study, has shared decision-making (SDM) as an intergraded concept in the deprescribing of medications to patients in end-of-life care [28,36]. SDM focuses on patient problems and allows the patient to decide on the intervention equally with the clinician [34,37]. When patients engage in SDM, they become more aware about their value and feel more informed. Some patients might indicate a preference to continue a PIM. In this context, while it can be helpful to understand the reasons underlying this preference, it seems that the great majority of patients would be willing to deprescribe one or more medication if their pharmacist and/or physician indicated it was appropriate and possible [28,34,36]. Altogether, SDM appears to have major potential in deprescribing within everyday clinical work. 

In the future, electronic deprescribing tools are needed to easily identify patients with medications suitable for deprescribing, thus helping to free up important time for clinicians. This must be seen in the light of the fact that manual screening is time-consuming and that deprescribing has to be a practical part of everyday clinical work. Furthermore, we require more knowledge about which patients have the best effect of medication deprescribing in relation to readmission and mortality. Recent literature has shown that the use of soluble urokinase-type plasminogen activator receptor (suPAR), as a non-specific inflammatory marker is strongly associated with disease burden, disease progression, readmission, and mortality [38,39,40,41,42]. It would be of high scientific value to investigate the effect of a deprescribing intervention in patients with low, intermediate or high suPAR levels. 

### 3.3. Strengths and Limitations 

The main strength of this study is that the intervention is integrated into daily clinical practice. An additional strength is the study design (RCT) and the use of blinded evaluators to review the deprescribing suggestions. This study also has several limitations. First, as this was a pilot study investigating feasibility, it was not designed to assess the clinical effect of intervention. Second, this was a single-center study with a single intervention time point. Third, our results were unable to assess whether patients’ total number of medications changed at 365+ days post inclusion. However, it is likely that the intervention had an impact on that endpoint as 97% of all deprescribed medications were sustained after 365+ days post intervention. If the total number of medications was to be investigated as an endpoint, it would be necessary to account for disease progression between the intervention and control groups over time. Fourth, we did not include quality of life (QOL) as an endpoint in this study. A recent systematic review by Pruskowski et al. found significant improvement in QOL in only 2 out of 10 deprescribing studies that used QOL as a primary endpoint. The authors concluded that deprescribing interventions probably do not increase QOL but that there is lack of well-designed studies in the field to definitively answer this question [43]. In our study, deprescribing was performed due to side effects in 8 out of 34 patients (24%), which means that measurement of QOL could have been relevant. Finally, we do not know whether our intervention is cost-effective. To address these limitations, future efficacy studies should be performed over longer time periods with multiple follow-up timepoints and multiple different endpoints. 

## 4. Materials and Methods 

### 4.1. Ethics Approval and Trial Registration 

The present study was conducted in accordance with the Declaration of Helsinki. All patients included in the study received a written informed consent at inclusion. The study was approved by the Danish Data Protection Agency (VD-2019-09) and registered at Clinical Trials.gov (identifier: NCT03912103).

### 4.2. Setting 

In 2016, the public Amager Hospital (part of Amager and Hvidovre Hospital group of the Capital Region of Denmark) had 16,958 visits in the acute setting. Of these visits, 13,851 (82%) were medical patients. Out of these, 1595 (12%) patients were referred to an outpatient clinic, either at Amager or Hvidovre Hospital. Amager Hospital has pulmonary, endocrinology, and cardiovascular outpatient clinics, representing three of the major specialties surrounding multimorbidity. In 2018, a SMOC was established at Amager University Hospital. SMOC is for multimorbid patients with severe unspecific symptoms and was established to reduce the number of acute hospitalizations. The multimorbid patients were referred by their own GPs or by an acute community team nurse. In the SMOC, the patients are accessed by a senior physician and a nurse. Additionally, in 2019, a clinical pharmacist was introduced to the SMOC staff team in order to evaluate the potential value of a collaborative medication review focusing on deprescribing to multimorbid patients. 

The clinical pharmacists were available in the SMOC between 08:00 and 15:00 Monday to Friday. Pharmacist duties included recording an accurate and complete medication history, supporting medication reconciliation, performing a patient-centered dialogue about patients’ medications and symptoms, giving recommendations for medication adjustment and deprescribing, and providing follow-up on any medication changes. In the SMOC, patients were first consulted by a nurse who performed clinical measurements (blood tests, blood pressure, etc.). The patients were then assessed by a clinical pharmacist who performed a patient-centered interview about the patient’s medications. The clinical pharmacist prepared for this interview based on data in the electronic patient record as well as any relevant results from clinical measurements. Next, each patient’s case was discussed by a multidisciplinary team. A physician then completed a patient exam to decide which medication changes should be implemented, and these changes were communicated to the patient by a clinical pharmacist. Finally, the pharmacist planned a follow-up call to discuss the status of medication changes, including deprescribing.

### 4.3. Trial Design and Participants 

This was a pilot RCT with multimorbid adult patients referred to the SMOC. Patients were enrolled in this study from March 2019 to December 2019. The inclusion criteria were (i) age ≥18 years and (II) multimorbidity (≥2 chronic diagnoses). The exclusion criteria were (i) inability to understand Danish, (ii) inability to cooperate cognitively, and (iii) hospitalization directly from the SMOC or up to 14 days after study inclusion. Patients were screened on weekdays based on a computer-generated list that included all patients meeting the inclusion criteria. 

### 4.4. Baseline Data Collection 

Demographic information was registered at inclusion. Data regarding diagnoses and baseline data were obtained from the patient’s electronic health records (Sundhedsplatformen, Epic Systems Corporation) as well as by patient interview.

### 4.5. Best Possible Medication List and Medication Reconciliation 

A complete medication list for each patient was prepared by a clinical pharmacist using the patient’s interview and the Shared Medication Card (SMC), a central database containing information for all Danish citizens regarding medications prescribed within the previous two years [43]. This medication list was then recorded in the electronic patient record and approved by a physician.

### 4.6. Intervention 

On the day of inclusion, a structured deprescribing intervention was performed by a clinical pharmacist in collaboration with the physician. Potentially inappropriate medications were identified using the STOPP criteria [44], the Danish deprescribing list [45], and relevant clinical guidelines. If a medication was identified as potentially inappropriate for any of the above reasons, the clinical pharmacist would recommend deprescribing that medication. These recommendations were then communicated to the patient during the patient interview. In identification of potential side effects, the medication was suggested for substitution or deprescribing. The pharmacist used elements from the motivational conversation tool to help the patients understand the potential benefits and risks from the proposed deprescribing. This tool was also used during the follow-up interviews with the patient. The medications the pharmacist had suggested to be deprescribed were discussed with the physician, who decided to accept, reject, or alter the recommendation. All deprescribed medications were documented in the electronic patient record by a physician and communicated to the patient’s GP.

#### 4.6.1. Telephone Consultation Related to Described Medications Seven Days after Intervention 

Seven days after visiting the SMOC, a telephone call took place between the pharmacist and the patients in the intervention group. The goals of this telephone call were to (i) clarify whether the patient had any questions relating to the deprescribed medication(s) and (ii) assess whether the patients had any unwanted health issues related to the deprescribing. If any health-related problems were identified by the pharmacist, the chief physician was contacted. 

#### 4.6.2. Telephone Data Collected 30 Days Post Inclusion

For all patients included in this study, the clinical pharmacist obtained a new updated medication list 30 days after inclusion. This medication list was prepared by the clinical pharmacist using the SMC, combined with the information provided by the patient during the telephone interview. 

#### 4.6.3. Telephone Data Collected 365+ Days Post Inclusion

All patients with ≥1 deprescribed medications sustained at 30 days post inclusion were reinterviewed by the clinical pharmacist in January 2021 (median follow-up time 550 ± 110 days) to determine whether the deprescribed medication was sustained 365+ days post inclusion. Information provided by the patient was cross-referenced with the SMC to determine whether each recommendation was sustained. This outcome was not pre-specified in the trial registry but added afterward following a suggestion made in the peer review process. The reason for this expansion was to investigate the long-term status of any medications that were deprescribed during the study and to see if they remained deprescribed.

#### 4.6.4. Acute Admission after Inclusion

For all patients included in this study, data regarding any acute admission 30, 90, and 180 post inclusion were collected from the patient’s electronic health records (Sundhedsplatformen, Epic Systems Corporation). 

### 4.7. Outcomes

The primary outcome of the study was to determine the feasibility of a collaborative medication review intervention. This was determined by measuring the differences in deprescribing rates for patients in the intervention versus control group with ≥1 medication deprescribed with a sustained deprescribed status 30 days after inclusion. Secondary outcomes were (i) change in total number of medications between the groups from inclusion during the SMOC visit versus 30 days after, (ii) percentage of eligible patients that agreed to participate in the study, (iii) percentage of patients who completed the study, (iv) percentage of deprescribed medications sustained at 365+ days in the intervention and control groups. 

#### Assessment of the Primary Outcome

The deprescribing status at 30 days post inclusion was assessed by comparing medication lists obtained at inclusion in the SMOC and at day 30 over telephone. Two evaluators blinded for group assignment independently reviewed whether the deprescribing suggested in the SMOC was implemented 30 days post inclusion. Any deprescribing discrepancies found between the evaluators were reviewed in person and consensus was reached. For cases where there was a lack of information, the participant was contacted again by telephone (if necessary) or the relevant information was accessed via the electronic patient record.

### 4.8. Sample Size 

This study was conceived as a pilot RCT. The primary outcome of interest was thus feasibility of the intervention. A formal power calculation was undertaken to demonstrate feasibility. The goal was 72 participants completing the study period of 30 days, including an expected dropout rate of 15%.

### 4.9. Randomization 

Patients were randomized 1:1 to intervention or standard care. The randomization sequence was determined by an independently generated random numbers table using RStudio 3.2.3. (Integrated Development for R. RStudio, Inc., Boston, MA, USA.). The random numbers table was kept by a secretary external to the study and was unavailable to us in order to maintain blinding. The external secretary assigned participants to groups using a sealed envelope system. Group allocation was concealed from the research clinical pharmacist, research physician, and participants until baseline data, and a possible medication list with medication reconciliation was obtained.

### 4.10. Statistical Methods

All patient characteristics are presented as means with standard deviation (SD). Differences in proportions between patient groups were assessed with chi-square test. Fisher’s exact test was used when the expected cell frequencies were lower than 5. Differences in patients’ total numbers of medication between time points were analyzed with a paired *t*-test. For all statistical tests, *p* < 0.05 was considered statistically significant. All calculations and statistical analyses were performed in RStudio 3.2.3. (Integrated Development for R. RStudio, Inc., Boston, MA, USA).

## 5. Conclusions

Among multimorbid patients with polypharmacy in a SMOC, a collaborative deprescribing intervention by the hospital pharmacist and physician resulted in significantly more patients with ≥1 sustained deprescribed medications compared to the control group (56% vs. 12%). On average, 1.1 and 0.2 medications were deprescribed per patient in the intervention and control groups, respectively. The total number of medications per patient was significant lower in the intervention group 30 days after intervention, while it remained unchanged in the control group. At 30 days post intervention, the status for 90% of the deprescribed medications remained as deprescribed. Importantly, at 365+ days post intervention, the deprescribing status of 97% of the deprescribed medication was also sustained from 30 days post inclusion. Of the eligible patients, 95% were included in the study. Of these, 93% completed the study. Altogether, our results indicate that a collaborative deprescribing intervention in a SMOC is feasible. Future studies should investigate the effect of such interventions in RCTs to identify which subgroups of patients benefit most. Finally, we recommend that policymakers, GPs, and especially hospital pharmacies be aware of the strategic possibilities of including SMOCs in the fight against inappropriate polypharmacy.

## Figures and Tables

**Table 1 metabolites-11-00204-t001:** Baseline characteristics of the included patients.

Parameter	Control Group (*n* = 33)	Intervention Group (*n* =34)	*p* Value	Total (*n* = 67)
Sex (men), *n* (%)	18 (55)	20 (59)	0.35	38 (57)
Age (years), mean (SD)	73.3 (10.3)	71.8 (14.2)	0.91	72.5 (12.3)
Number of drugs, mean (SD)	10.5 (4.0)	9.3 (3.2)	0.18	9.9 (3.7)
Regular drugs, mean (SD)	8.8 (3.4)	7.3 (3.1)	0.063	8.0 (3.3)
Pro re nata drugs, mean (SD)	1.9 (1.4)	2.0 (1.5)	0.78	1.9 (1.5)
eGFR (mL/min/1.73m^2^), mean (SD)	66.2 (18.5)	68.9 (18.4)	0.55	67.6 (18.4)
Comorbidities, mean (SD)	4.5 (1.4)	4.3 (1.4)	0.56	4.4 (1.4)
Acute admission within 30 days, *n* (%) Acute admission within 90 days, *n* (%) Acute admission within 180 days, *n* (%)	8 (23.2) 14 (42.4) 21 (63.6)	3 (8.8) 9 (26.5) 13 (38.2)	0.11 0.20 0.052	11 (16.4) 25 (34.3) 34 (50.7)

^1^ SD, standard deviation; eGFR, estimated glomerular filtration rate.

**Table 2 metabolites-11-00204-t002:** Number of patients with sustained deprescribed medications from inclusion to 30 days post inclusion.

	Control Group (*n* = 33)	Intervention Group (*n* =34)	*p* Value
Number of patients with ≥1 medication deprescribed, *n* (%)	4 (12)	19 (56)	<0.001
Number of patients with ≥2 medication deprescribed, *n* (%)	1 (3)	9 (26)	0.007
Number of patients with ≥3 medication deprescribed, n (%)	0 (0)	4 (12)	0.042

**Table 3 metabolites-11-00204-t003:** Most frequently deprescribed medications in the intervention group.

Class and Medication	Frequency *n* (%)
Analgesics Paracetamol NSAID Gabapentin Cardiovascular Beta blockers Antihypertensives Statins Vasodilators Gastrointestinal Proton pump inhibitors (PPIs) Stimulant laxatives Sedatives * Others Total	14 (37.8) 5 (13.5) 4 (10.8) 2 (5.4) 9 (24.3) 4 (10.8) 2 (5.4) 1 (2.7) 1 (2.7) 5 (13.5) 3 (8.1) 2 (5.4) 3 (8.1) 6 (16.2) 37 (100)

* Sedatives include nonbenzodiazepines, benzodiazepines, and first-generation antihistamines.

## Data Availability

Data available on request due to restrictions. The data presented in this study are not publicly available due to Danish legislation. Request to access the dataset will require an individual inquiry to the Danish Data Protection agency for approval.

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
