# Peer review of "A Collaborative Deprescribing Intervention in a Subacute Medical Outpatient Clinic: A Pilot Randomized Controlled Trial"

_metabolites, 2021, doi:10.3390/metabo11040204_

Round 1

Reviewer 1 Report

An interesting pilot study on an important topic. I agree with the cited limitations of this study, but as a pilot study it is appropriate --additional time points and actual effects of the intervention would be good to explore further in future studies. For this paper, the role and specific duties of the clinical pharmacist in the MOC could be a bit better defined so that other clinics wishing to implement similar programs had more detail in this regard. How does this individual interact with patients and staff on a daily basis? Are other pharmacy personnel involved? Perhaps a figure that shows some STOPP tool criteria related to certain potentially inappropriate medications would be helpful for the reader as well.   

Minor issues in wording and style were detected, for example, line 69 says "...through i.e. reduction of drug-related problems." Say either "through" OR "i.e." but not both. Similarly, in the abstract, line 21 should say "inappropriate medication use in multimorbid..." Line 23 "The objective of our pilot study..." Minor editorial changes to grammar and wording would strengthen the manuscript. 

Author Response

Please see the attached PDF file.

Reviewer 2 Report

The study highlights the feasibility of pharmacist led deprescription in MOC-SCU. Major improvement of the study design should be considered before the manuscript is accepted. 

  1. Although the authors indicate that this is a pilot study, I believe that a follow up of only 30 days is not sufficient to make conclusion about the significance of the results considering the study was conducted for 10 months in 2019.
  2. Another reason for the above-mentioned comment is that the authors indicate in line 175 was a study that found no difference in the total number of medications prior to interventions and during follow up. A longer follow up will indicate the sustainability of the interventions, which I believe is important.
  3. While polypharmacy is an issue, the authors should also focus a bit more on the improvement of the quality of life on the patients and potential side effects they were facing. For successful interventions, describing this will make the study more significant.
  4. Although the authors used similar methods in their study, a lot of the methods are written in using same sentences as reference 16 (which has some of the authors in this study) which can be considered self-plaigiarism.
  5. The novelty of this study is a bit weak as the authors performed the same study as they did previously (as reference 16).

To be considered for publication, I believe the authors should consider a longer follow up.

Author Response

Please see the attached PDF file.

Reviewer 3 Report

The paper is very well written and easy to read. The study design (RCT) is appropriate. Results of the feasibility and sustainability of pharmacist led deprescribing are very interesting. Conclusions of the authors are in adequation with the results. Strengths and limits a presented and discussed. 

Even with a feasibility objective, the paper address a very important concern with medication use and deprescription and highlight the crucial rôle of multidisciplinary teams including clinical pharmacist to optimize patients pharmacotherapy. 

I strongly recommend publication. 

Author Response

None

Reviewer 4 Report

Thank you for the submission.  However, the content seems well out of scope for Metabolites. Irrespective, I feel that the manuscript does not have sufficient importance and novelty for international publication. It would add nothing new to the existing extensive literature on the same topic.  It is a very small feasibility study. The rates of deprescribing recommendations and implementation were very low and somewhat surprising given the average number of total medications was 10, with very little change observed in the overall use of medicines as a result. One could actually question the value of the service based on this very modest outcome; there has in fact been minimal deprescribing achieved. There are no clinical outcomes recorded. The introduction could be more concise.

Reviewer 5 Report

 Here they are my comments:

Abstract

You state that 'multidisciplinary', but your study is focused on 'physician' and 'pharmacist'. How about 'nurses' and other persons involved in medicines management? If only these 2 professionals have been involved, please remove multidiscplinary word from throughout your article. If not, the role of other members should be described and specified also.

Both the research design and the data collection tool should be recognised here.

Introduction

In some places you have defended the pharmacy-led medication process would suits the patient, but later you discuss about the multidisciplinary perspective to medication. With regard to the significance of pharmacy-led initiatives, the following citations should be used:

https://www.mdpi.com/2226-4787/7/3/128

https://www.mdpi.com/2226-4787/7/3/82

https://www.mdpi.com/2226-4787/7/2/56

https://www.mdpi.com/2226-4787/6/1/2

On the other hand, the multidisciplinaty team involved in medication (healthcare professionals and patients) should be clearly specified and their role should be described. How the multidisciplinary identity is going to be used in your research process?

Methods

I can see that 'methods' section is a missed part of this article. You need to describe in details the research process using the following subheadings: Research design, Sample and setting, Intervention (procedure), Data collection, Data analysis, Ethical considerations. 

These details are needed to make it possible to repeat your study in other settings with a similar workflow and condition. 

Also, you are advised to use the Equator apropriate to your research process to write your article: https://www.equator-network.org/

Results

It consists of the details of the medication process in the outpatient clinic, but the phrmacy-led initiatives/roles in the multidiscilinary context are unclear. You need to present the results in connection with the intervention made in the setting and the research aim.

Discussion

This needs to be improved in terms of the multidisciplinary idenity of the research process. 

Also, some findings have been presented in the discussion that can not be traced back to the Results section. For instance, Line 227. Please ensure that all findings have been presented in the results section and mutually all have been discussed in the discussion section.

Lines 264 and 267 contradicts each other! Is this a RCT or not? Is this a strenght or limitation?

Why Materials and Methods have been placed after the Discussion?! Check that the details matches the required subheadings (see the above comment on the Methods) and use the Equator checklist, also.

Conclusion

It should contain practical strategies in terms of policy-making, management, practice and education for improving the current condition. 

Author Response

Please see the attached PDF file.

Round 2

Reviewer 2 Report

The authors have addressed all questions raised.

Author Response

Thanks. Your comments have increased the quality of our manuscript.

Reviewer 4 Report

Thank you for the revision. However, I still believe that the manuscript does not have sufficient importance and novelty for international publication. It would add nothing new to the existing extensive literature on the same topic. It is a very small feasibility/pilot study that does not warrant international publication. There have been much larger full trials and I see no reason for publishing a tiny pilot study on the same topic. The methods are absent/superficial e.g. how was randomisation performed? Also, the rates of deprescribing recommendations and implementation were very low and somewhat surprising given the average number of total medications was 10, with very little change observed in the overall use of medicines as a result. One could actually question the value of the service based on this very modest outcome; there has in fact been minimal deprescribing achieved. There are no clinical outcomes recorded.

Author Response

Please see the attached PDF file.

Reviewer 5 Report

Nothing more.